# Remdesivir—Bringing Hope for COVID-19 Treatment

**Naser F. Al-Tannak [1,*,†], Ladislav Novotny [1,†]**  **and Adel Alhunayan [2,†]**

[1]  Department of Pharmaceutical Chemistry, Faculty of Pharmacy, Kuwait University, P.O. Box 24923, Safat 13110, Kuwait; novotny@HSC.EDU.KW

[2]  School of Medicine, Health Science Center, Kuwait University, P.O. Box 24923, Safat 13110, Kuwait; alhunayan@HSC.EDU.KW

*  Correspondence: dr_altannak@hsc.edu.kw

†  All authors contributed equally to this work.

**Abstract:** At the beginning of 2020, the world was swept with a wave of a new coronavirus disease, named COVID-19 by the World Health Organization (WHO 2). The causative agent of this infection is the severe acute respiratory syndrome coronavirus 2 (SARS-CoV-2). The data available on one of the promising therapeutic agents—nucleotide analog remdesivir (Gilead Sciences number GS-5734)—were evaluated. These data were concerned with remdesivir activation from the prodrug to the active molecule—triphosphate containing 1'-cyano group and modified nucleobase. This triphosphate competes with the natural substrate adenosine triphosphate. Additionally, its mechanisms of action based on RNA and proofreading exonuclease inhibition, leading to the delayed RNA chain termination of infected cells, and basic pharmacological data were assessed. Additionally, the analytical determination of remdesivir and its metabolites in cells and body liquids and also some data from remdesivir use in other RNA infections—such as Ebola, Nipah virus infection, and Middle East Respiratory Syndrome (MERS)—were summarized. More recent and more detailed data on the clinical use of remdesivir in COVID-19 were reported, showing the intensive efforts of clinicians and scientists to develop a cure for this new disease. Remdesivir as such represents one of the more promising alternatives for COVID-19 therapy, however the current understanding of this disease and the possible ways of dealing with it requires further investigation.

**Keywords:** remdesivir; GS-5734; COVID-19; WHO; RNA-dependent RNA polymerase; endonuclease; RNA viral infections; GS-441524

## 1. Introduction

The year 2020 will be remembered, as at its beginning, a new disease, COVID-19, that is caused by the severe acute respiratory syndrome coronavirus 2 (SARS-CoV-2), affected people around the world [1]. What was first reported as a local event in a city, Wuhan, in China on 31 December 2019 became a Public Health Emergency of International Concern on 30 January 2020, with the first case of this pneumonia reported outside of China in Thailand on 13 January 2020. China confirmed COVID-19 transmission between humans on 20 January. This general development led the World Health Organization (WHO) to declare the infection a pandemic on 11 March 2020 [1]. In the second half of May 2020, over 4.5 million confirmed cases of COVID-19 have been registered around the world, and at least 310,000 COVID-19 patients have succumbed to this disease [2].

As of May 2020, there is a frantic search for suitable therapeutic means to cure this disease. Various drugs have been repurposed for this reason with more or less success. These include lopinavir/ritonavir, favipiravir, darunavir/cobicistat, camostat mesilate/nafamostat, tocilizumab, chloroquine/hydroxychloroquine, colchicine, baricitinib, aviptadil, eculizumab, and remdesivir [3]. As remdesivir is a very promising nucleoside analog originally developed for the treatment of Ebola

disease, the goal of this review is to summarize the current knowledge about remdesivir and its COVID-19 effects.

## 2. Method

This review is based on selective literature searches in PubMed, with remdesivir being always one of the words that any search was based on. There were over 100 articles on this topic found on PubMed in May 2020. Other public domain documents and web pages (i.e., from the WHO, CNN, PubChem, etc.) were also consulted.

## 3. Remdesivir as a Chemical Molecule and a Prodrug

Remdesivir was developed as an antiviral agent by the company Gilead Sciences under the name GS-5734 [3]. Remdesivir is a prodrug requiring bioactivation within cells. Remdesivir is a nucleotide analog (Figure 1) with a molecular weight 602.585 g·mol$^{-1}$ and cumulative formula $C_{27}H_{35}N_6O_8P$. The IUPAC (International Union of Pure and Applied Chemistry) name for remdesivir is 2-ethylbutyl (2$S$)-2-[[[(2$R$,3$S$,4$R$,5$R$)-5-(4-aminopyrrolo [2,1-f][1,2,4]triazin-7-yl)-5-cyano-3,4-dihydroxyoxolan-2-yl] methoxy-phenoxyphosphoryl]amino]propanoate (Figure 1) [4].

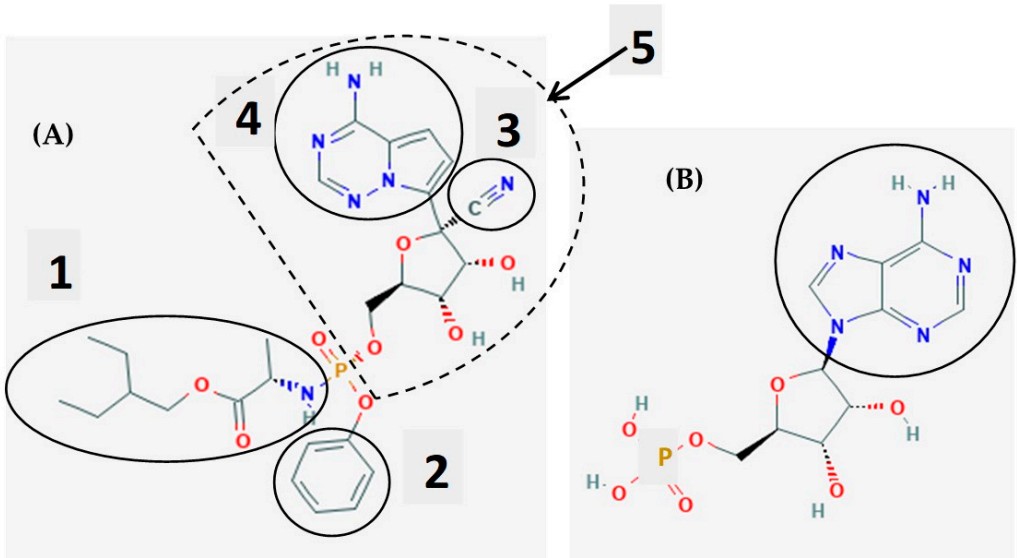

**Figure 1.** The chemical formula of remdesivir (**A**), adenosine 5′-phosphate (**B**), and of GS-41524 within A [4]. Features of remdesivir molecule: (**A**) (1) Amido substituent of the phosphoric acid moiety. It is removed during remdesivir activation within cells. (2) Phenyl substituent of the phosphoric acid moiety. It also is removed during remdesivir activation within cells. (3) Cyano group at carbon 1 of ribose. Cyano group is not removed during activation and is an important molecular feature contributing to the activity of the remdesivir active metabolite—a triphosphate of nucleoside GS-441524. (4) A modified nucleobase of remdesivir (**A**)—compare to the formula of adenosine 5′-monophosphate. (**B**) According to the IUPAC nomenclature, the name of the nucleobase substituent is 4-aminopyrrolo[2,1-f][1,2,4]triazin-7-yl-. (5) GS-441524—an active nucleoside arising from remdesivir that requires phosphorylation to its triphosphate to exhibit its activity.

There are several basic molecular features of remdesivir, which is an adenosine nucleotide analog: (1) As an adenosine nucleotide analog, it compromises RNA synthesis (and not DNA synthesis), and consequently it is active against RNA viruses. (2) The modification of the phosphate group by the substituted amido group increases the lipophilicity of this molecule and forms a prodrug that needs to be activated in the active molecule within the cell through an alanine-substituted intermediate; additionally, the phosphate is also modified by substitution with a phenyl (phenoxy) group that is

also increasing lipophilicity (see Figure 1). (3) Cyano group at position 1 of ribose of remdesivir and its metabolites and the modified nucleobase of this synthetic nucleoside analog are essential for the disruption of RNA synthesis (Figure 1). Gilead Sciences has given the nucleoside with the cyano substitution and a modified nucleobase the number GS-441524 [4]. The active molecule of remdesivir in cells is a triphosphate, with its most important activity being the inhibition of RNA-dependent RNA polymerase [3,5]. The active triphosphate is in fact a triphosphate of GS-441524 (C-nucleoside, with ribose connected to the nucleobase via a bond formed between two carbons) that is a final product of the remdesivir activation. However, the activation is a more complex process involving an esterase that removes the ester of 2-ethylbutanol and 2-aminopropanoic acid in the lipophilic part of remdesivir. This is followed by an intramolecular rearrangement, resulting in alanine residue connected to the phosphoric acid moiety via nitrogen (thus forming phosphoamide). This is consequently attacked by a phosphoamidase-forming nucleoside monophosphate that is finally activated to the triphosphate by nucleoside-phosphate kinase. The nucleoside monophosphate may be degraded to a nucleoside that, in turn, can be again re-phosphorylated by kinases [6].

## 4. Mechanisms of Activity of Remdesivir

The main mechanism of the activity of remdesivir is based on the similarity of its active metabolite triphosphate (containing a cyano group and a modified nucleobase—4-aminopyrrolo[2,1-f] [1,2,4]triazin-7-yl-) to a natural substrate necessary for RNA synthesis—adenosine triphosphate. The active molecule affects the activity of viral RNA-dependent RNA polymerase in COVID-19-infected cells (or other cells infected by one of the RNA viruses). Additionally, it compromises the work of exoribonuclease responsible for the proofreading of newly synthesized RNA. This leads to a decrease in the production of viral RNA [3,5]. Remdesivir is classified as a delayed chain terminator [6] as its induction, an irreversible chain termination, is not immediate, but the RNA chain termination takes place after an additional five nucleobases are incorporated into an RNA chain that is being synthesized [7]. Other work reports the incorporation of only three additional nucleotides after the incorporation of the active triphosphate of remdesivir [8]. Potential resistance to remdesivir was already identified that would make its therapeutic use less effective [9]. However, resistance to remdesivir is overcome with increased but nontoxic concentrations of this nucleotide analog [10]. An additional important aspect of remdesivir effectivity in fighting viral corona infections it that its activity is of a broad-spectrum. Remdesivir inhibits both epidemic and zoonotic coronaviruses [11]. When evaluating remdesivir for its effectivity in COVID-19, it is necessary to take into account the presence of the exonuclease domain in the structure of this virus. The role of this exonuclease is to excise nucleotide antimetabolites that should not be incorporated into the RNA. This may play an important role in the development of resistance to nucleoside-based therapeutics. Remdesivir inhibits active sites in both COVID-19 enzymes (RNA-dependent RNA polymerase and exonuclease). This leads to a high incorporation, delayed chain termination, and decreased excision. All of these effects are caused by the presence of a cyano substituent at position 1' of the ribose and modified nucleobase (Figure 1). Remdesivir in infected cell possesses (due to its inhibition of both enzymes) a good balance between incorporation into RNA molecules and the excision of the incorporated nucleotide analog (Figure 2) [12].

A molecule transformer-drug target interaction model determined the interaction of remdesivir and some other drugs with the viral proteins of COVID-19. Remdesivir was found to be the second-best compared to atazanavir, which is used as an antiretroviral medication in HIV patients. The inhibitory potency against the COVID-19 3C-like proteinase (chymotrypsin-like protease or 3CL$^{pro}$) demonstrated by atazanavir was a $K_d$ of 94.94 nM and by remdesivir was 113.13 nM. The values of the other drugs investigated were less favorable [13]. These findings were also confirmed by the in silico docking models [14] and the integrated computational approach [15]. Currently, a model of the RNA-dependent RNA polymerase of COVID-19 is also available which improves the in silico testing of various potential drugs against this viral infection [16].

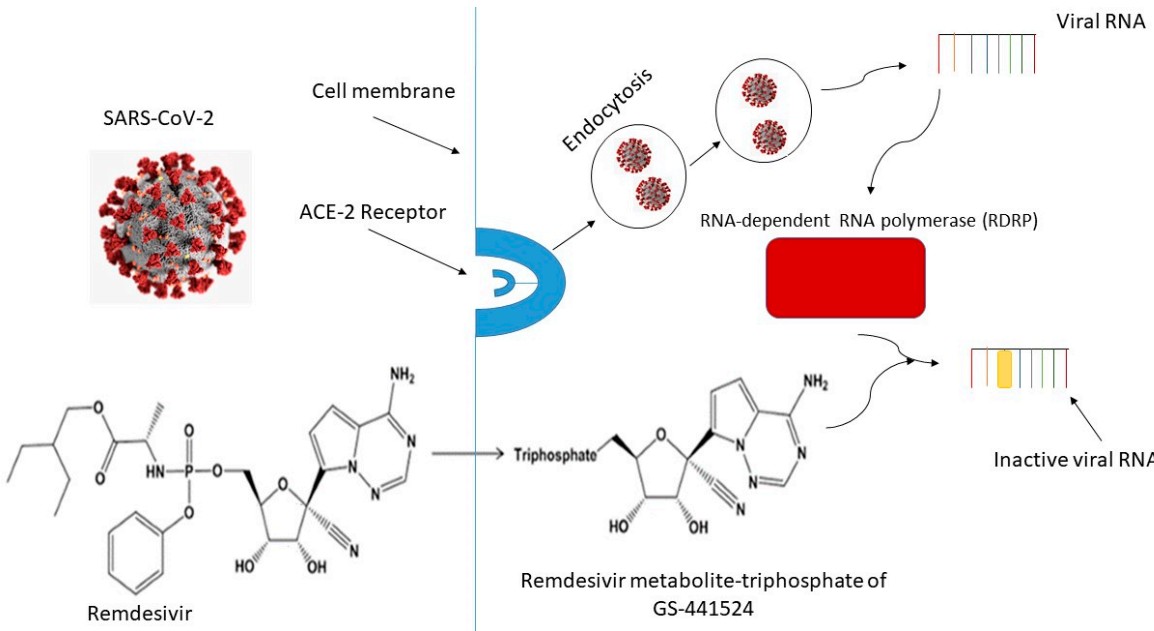

**Figure 2.** Remdesivir mode of action in inhibiting the RNA of the virus from replication in an infected cell.

## 5. Basic Pharmacological Data of Remdesivir and Its Adverse Effects in Patients

After intravenous administration, the plasma half-life of remdesivir in the primate model is 20 min. However, the active molecule—triphosphate with 1'-cyano group and 4-aminopyrrolo [2,1-f][1,2,4]triazin-7-yl- subtiutent as a nucleobase—is more stable within the cellular environment [6]. The half-life time of the active triphosphate in non-human primates is 14 h and it is 20 h in humans [17].

Remdesivir at the dose of 10 mg/kg was shown to be distributed to the testes, epididymis, eyes, and brain within 4 h [17]. However, data on the remdesivir routes of elimination, volume distribution, clearance, elimination, protein binding, and overdose are not available yet.

A randomized, double-blind, placebo-controlled, multicenter trial (registered in the database ClinicalTrials.gov under the number NCT04257656) evaluating the effect of remdesivir (200 mg on day 1, followed by 100 mg on days 2–10 in single daily infusions) has shown that the adverse effects of this therapy observed were typically respiratory failure; organ impairment, as indicated by low albumin; low potassium; and low red blood and platelet cell counts. A yellow coloration of the skin was also observed [18]. Other reports show the occurrence of gastrointestinal distress, elevated transaminases, infusion site reactions [19], low blood pressure, nausea, vomiting, sweating, and shivering [20].

## 6. Analytical Determinations of Remdesivir, Its Metabolite GS-441524, and Its Triphosphate

Therapeutic drug monitoring, which is concerned with the measurement of drug concentrations in biological fluids to optimize the drug regimen and evade toxic properties or therapeutic letdowns, is already well known in numerous areas, such as in HIV treatment, and may be valuable in the issue of COVID-19 therapy [21–23]. Therefore, both for pharmacokinetics studies and probable future therapeutic drug monitoring, there is an urgent need for accurate and precise analytical methods to quantify remdesivir and/or its metabolite GS-441524 in human plasma. Nevertheless, in literature the information about the pharmacokinetics and pharmacodynamics of remdesivir in humans is insufficient, as well as no therapeutic and toxic ranges have been reported. The reason for such insufficiency is the few cases treated with remdesivir. Surprisingly, a remdesivir qualitative and quantitative analysis was discussed only in a very small number of studies. In the first study, Avataneo and her colleagues [24] studied the analysis of remdesivir and its metabolite GS-441524 in human plasma by using the UHPLC-MS-MS technique. Their research team used the UHPLC system

coupled with a Triple Quadrupole for a chromatographic analysis. A chromatographic separation was obtained on an Acquity® HSS T3 1.8 µm, 2.1 × 50 mm column and a physical filter ('Frit', 0.2 µm, 2.1 mm) with a pre-column set at 40 °C by using a column thermostat. The optimum separation was achieved by using a mobile phase consisting of H2O + 0.05% formic acid and acetonitrile +0.05% formic acid in gradient elution mode. 6,7-dimethyl-2,3-di(2-pyridyl) quinoxaline was used as an internal standard in this study. Positive electrospray ionization (ESI+) was used for all the analytes. Multiple reaction monitoring (MRM) traces (m/z) were quantified as 603.15 > 200 for remdesivir, 292,163 for GS-441524, and 313.2 > 78.05 for the internal standard. The rapid protein precipitation was performed using methanol: acetonitrile (50:50 *v/v*) for the extraction of remdesivir and its metabolite GS-441524. The established method was shown to be accurate, precise, sensitive, and linear. Moreover, the developed method is shown to have a lower limit of quantitation (LLOQ) for remdesivir, and GS-441524 was 0.98 ng/mL, while the LOD values were 0.24 ng/mL for remdesivir and 0.98 ng/mL for GS-441524. Both remdesivir and GS-441524 remained as stable in-stock solutions when stored at −80 °C for over 4 months. Additionally, remdesivir was revealed to be stable in the stock solution for at least 10 months, while the stability of the GS-441524 stock solution in the same conditions had not been tested yet. Although remdesivir was found to be unstable at room temperature and at 4 °C when dissolved in plasma for 24 h, remdesivir was stable for 7 days in the extracted plasma samples in the auto-sampler (10 °C). This method was not tested on real-life samples but will be very useful in studies of the pharmacokinetics of remdesivir [24]. In the second study, high-performance liquid chromatography was used to assess the remdesivir purity during the whole synthesis process. In this study, Kinetex® C$_{18}$ (2.6 µm, 100 × 4.6 mm) was used as a stationary phase, and 0.1% trifluoroacetic acid in water and 0.1% trifluoroacetic acid in acetonitrile was used as a mobile phase. The mobile phase was run in a gradient mode at a flow rate of 1.5 mL/min. Moreover, the progress of the reaction during synthesis was monitored by using LC-MS equipped with a Gemini® C$_{18}$ 5 µm × 30 × 4.6 mm column. The study pointed out the role of HPLC in evaluating remdesivir synthesis, screening, separation, and purification during the synthesis process as well as other antiviral drugs [25].

In 2018 Murphy and his colleagues examined remdesivir and its metabolite, C-nucleoside ribose analog, on domestic cats infected with feline infectious peritonitis (FIP) [26]. FIP is caused by a coronavirus that tends to attack the cells of the intestinal wall in cats. It was demonstrated that the remdesivir metabolite GS-441524 is effective, safe, and inhibits FIP virus replication. Murphy with his team evaluated the effectiveness, safety, and therapeutic doses of the nucleoside. During the study, the analysis of cells and body fluids for the concentration of C-nucleoside that is created from remdesivir and also for its triphosphate was performed in plasma, aqueous humor, and cerebrospinal fluid after precipitating the proteins by acetonitrile in the presence of an internal standard of 5-(2-aminopropyl)indole at the concentration of 20 nM. Following the filtration for protein removal, drying under a stream of nitrogen, and reconstitution with 0.2% formic acid and 1% acetonitrile, LC/MS-MS analyses for the GS-441524 concentrations were performed. The level of phosphorylation of GS-441524 to its triphosphate was determined in the frozen samples of cultured cells and peripheral blood mononuclear cells. The frozen cells were resuspended in 0.5 mL of 70% methanol with an internal standard 2-chloro-adenosine triphosphate (500 nM) and kept for 30 min at −80 °C. After drying and evaporating, the samples were reconstituted with an aqueous 1 mM ammonium phosphate. The pH of these reconstituted samples was 7.

A 50 × 2 mm, 2.5 µm Luna C18(2) HST column (Phenomenex, Torrance, CA, USA) connected to an LC-20ADXR (Shimadzu, Columbia, MD, USA) pump system and autosampler was used for the separation of analytes in a multi-stage linear gradient from 10% to 50% acetonitrile in a mobile phase containing 3 mM of ammonium formate (pH 5.0) with 10 mM of dimethylhexylamine at a flow rate of 150 µL/min. The MS/MS was operated in positive ion and multiple reaction-monitoring modes. The used instrumental method [26] seems to be very efficient and has enough precision.

## 7. Therapeutic Uses of Remdesivir

Remdesivir as an analog of ribonucleotide adenosine monophosphate has the potential to compromise RNA synthesis in viral RNA infections. It was tested against various RNA viruses with more or less success. Remdesivir was tested against Ebola, Nipah virus, and Middle East respiratory syndrome (MERS) in human medicine and feline infectious peritonitis. Currently, the world is following numerous clinical trials in which remdesivir is tested in patients with COVID-19.

### 7.1. The In Vitro and In Vivo Testing of Remdesivir

At the start of the COVID-19 pandemic, no specific therapy or specific preventive therapeutic agents were known and available. Various drugs were repurposed from other indications, and the information regarding their in vitro and in vivo activity in various cells, and also humans where from the time when these drugs were investigated against other infections [27]. In the case of remdesivir, Ebola infection use gave some information about its possible therapeutic activity (see Section 7.1.) as it was studied in vitro on its ability to be incorporated into the RNA structure, and on its ability to inhibit various RNA polymerases [7]. Additionally, it has already been studied in a non-human primate model with an Ebola infection [28,29]. Many studies on the remdesivir activation and mechanism of action were performed on cellular or sub-cellular models. The results of these studies are discussed in Sections 3 and 4. For example, Agostini and her team used in her study [10] murine astrocytoma cells and baby hamster kidney 21 cells expressing the murine hepatitis virus receptor. She also used the human lung epithelial cells Calu-3 and human tracheobronchial epithelial cells. As much as her data are interesting, they were not obtained specifically for SARS-CoV-2. Sheahan and his team [30] used in their experiment the primary human lung epithelial cell cultures. They studied remdesivir activity on circulating contemporary human CoVs—i.e., SARS-CoV, but not SARS-CoV-2. Only newer scientific reports deal with SARS-CoV. Choy [31] studied the activity of remdesivir and other substances against the SARS-CoV-2 virus in Vero E6 cells. A nice summary of remdesvir preclinical in vitro and in vivo data was published just at the end of May 2020 [32]. Another summarization of the in vitro data for remdesivir [33] and several other therapeutics used in COVID-19 appeared online also very recently in May 2020. As for in vivo studies, the preference is given to the use of primate models, as discussed in the sections on Ebola and MERS.

### 7.2. Ebola

Remdesivir was used in the treatment of Ebola cases based on preclinical data showing that it had blocked Ebola virus replication in primates [28,29]. It was used in Ebola patients in emergency settings during the Kivu Ebola epidemic in 2018–2019. Then, it was shown that remdesivir is inferior to the effectivity of monoclonal antibodies [29,34]. An excellent review of therapeutic strategies that may be useful against the Ebola infection was published only recently [30]. Similarly to the situation with other RNA viruses, remdesivir rivals adenosine triphosphate for incorporation into RNA. Kinetic experiments demonstrated that GS-441524triphosphate (the active molecule of remdesivir) is similar to adenosine triphosphate in its incorporation efficiency. However, the selectivity of this incorporation is approximately 4 times higher for adenosine triphosphate compared to GS-441524triphosphate [7]. On the other hand, human mitochondrial RNA polymerase effectively distinguishes these two substrates, and the selectivity ratio for adenosine triphosphate and GS-441524 triphosphate is approximately 500 [7]. The presence of activated remdesivir in cells results in delayed chain termination in the synthesis of RNA [7].

Remdesivir was used in two patients with Ebola successfully—both patients survived. This moved remdesivir into Phase 2 of clinical development for its use in Ebola patients [35,36]. Additionally, when used in a newborn from an Ebola virus-positive lady on the day of birth (together with monoclonal antibodies ZMapp), remdesivir contributed to the eradication of the virus in the child (as proved by PCR) and it seems to be doing well at the 12 months of age [37]. Despite these and some other positive

records, potential Ebola outbreaks still represent a danger for affected populations, especially due to the ways of transmission, various social aspects, and current limits of therapeutic interventions [38].

### 7.3. Nipah Virus

Nipah virus (*Paramyxoviridae*) is another pathogenic RNA virus. This virus was transmitted to humans from fruit bats. Additionally, transmission between persons was confirmed. Similarly to COVID-19 infection, Nipah virus also causes respiratory and neurological disorders, with a 40% to 75% mortality according to the World Health Organization [39]. Treatment, similarly to the treatment of the COVID-19 infection, is still under development. Remdesivir represents a viable option, as it seems to be highly effective in the African green monkey model [40]. Experiments performed using this model with infecting the experimental animals with a lethal dose of virus have shown that remdesivir administered in one dose daily during 12 days saved all the treated animals compared to the monkeys that were not treated. All the animals left without treatment died during the experiment [40]. It indicates that remdesivir may represent a suitable treatment also for the Nipah virus.

### 7.4. Middle East Respiratory Syndrome (MERS)

Coronaviruses belong to the Orthocoronavirinae family and transmission between various species is one of their properties. At present, therapeutic strategies for these viruses do not exist, but the present crisis has made researchers and medical doctors pay increasing attention to this issue. Remdesivir seems to represent a viable option in the treatment of various corona (and other) infections through the inhibition of the viral RNA dependent RNA polymerase [41]. Middle East Respiratory Syndrome coronavirus (MERS-CoV) serves as the causative agent of respiratory disease that has caused over 2468 infections in humans, with more than 851 deaths registered in 27 states during the last 8 years [30]. MERS-CoV infection seems to respond to the remdesivir therapy, as shown in the nonhuman primate model thorough the inhibition of viral replication, as demonstrated in vitro [42]. Treatment by remdesivir (starting 24 h before infection by MERS-CoV) resulted in the full inhibition of clinical symptoms by inhibiting MERS-CoV replication in the lungs, thus preventing the appearance of lung tissue lesions. The treatment with remdesivir in the early stages of MERS infection has shown significant clinical benefit, accompanied by the diminished formation and severity of lung lesions [42]. Compared to other agents (lopinavir, ritonavir, and possibly interferon beta), remdesivir shows the highest anti-MERS activity in vitro, and it improves various pulmonary functions in a murine model in vivo [30]. This effect is achieved through the competition of the triphosphate of activated remdesivir (1′-cyano analog of adenosine with adenine replaced by 4-aminopyrrolo[2,1-f][1,2,4]triazin-7-yl-) competing with ATP via various mechanisms [43].

### 7.5. COVID-19—Clinical Trials Evaluating Remdesivir as a Treatment for COVID-19

Originally, within the period of January to March 2020, remdesivir was given to individual COVID-19 patients on a compassionate-use basis. Remdesivir was administered for 10 days (200 mg intravenously on day 1, and 100 mg per day on the day 2–10). An analysis of data for 53 patients (22 in the USA, 22 in Europe or Canada, and 9 in Japan) has shown that 25 patients (47%) were treated successfully and 7 patients (13%) died. Clinical improvement was seen in 36 of 53 patients (68%) [44].

On 6 February 2020, the first global clinical trial began in China, specifically in Hubei. The designed clinical trial was a randomized, double-blind, placebo-controlled, and multi-center study, with a participation of 237 hospitalized patients (aged ≥ 18 years). The study found that there were no statistically significant clinical benefits associated with remdesivir use. However, the clinical improvement time for the patients was shorter while using the antiviral drug compared to those receiving placebo [18].

The first randomized and controlled clinical trial for remdesivir in the United States has been begun by the University of Nebraska Medical Center (UNMC) in Omaha to evaluate remdesivir's safety and efficacy in hospitalized adults diagnosed with COVID-19. The trial was overseen by the National Institute of Allergy and Infectious Diseases (NIAID), part of the National Institutes of

Health. The participants in the trial should be examined and diagnosed with SARS-CoV-2 infection as well as evidence of lung association, such as speedy sounds when breathing (rales), with a need for supplemental oxygen or abnormal chest X-rays, or barely breathing and requiring mechanical ventilation. Individuals with confirmed infection who have mild symptoms or no symptoms will not be involved in the study. The preliminary data analysis, as well as results, was released on 29 April 2020. The study showed that the recovery time was reduced by 31% for the patients who received remdesivir compared to the placebo group. Moreover, the mortality rate was 8.0% for the group receiving remdesivir, while it was 11% for the group who received a placebo [45].

Moreover, Gilead Sciences Inc. released data on a Gilead sponsored Phase 3 randomized trial in 12 years and older hospitalized patients with severe COVID-19 disease. The design of the study was randomized, open-label, and multicenter, and the estimated enrolment of the study was 6000 patients. The data revealed that patients who were treated for 5 days with remdesivir had a comparable clinical improvement compared to the patients who were treated for 10 days with remdesivir [45]. Additionally, the study indicated the proposed treatment duration for severe cases, which can aid in monitoring other COVID-19 cases that require treatment in an intensive care unit [45].

In the United States, a clinical trial phase 3 has been launched on 21 February 2020. The study is designed to be an adaptive, randomized, double-blind, placebo-controlled trial to assess the safety and efficacy of remdesivir in hospitalized adults (aged ≥ 18 years) diagnosed with COVID-19. The study was proposed to be a multicenter trial that will be conducted in roughly 100 sites all over the world. The study will compare different investigational therapeutic agents, including remdesivir, to a control. The project originally enrolled 394 patients, however, with the recent enrolment rates the total sample size could reach 600 or even more than 800. To date, the study results have not been posted or may not yet be posted because they are pending a quality control (QC) review by the National Library of Medicine (NLM) or the sponsor or investigator is addressing QC review comments provided by the NLM [46].

After all this, the clinical trials showed diverse results that were due to the different rules of patients' enrolment and different endpoints. The clinical trials performed in China had stricter conditions for the enrolment of patients, such as having an interval from symptom onset to the enrolment of 12 days or less. Besides this, the time (in days) from randomization to the point of a decline of two levels on a six-point ordinal scale of clinical status (1 = discharged and 6 = death) was the primary endpoint for the China trial. In contrast, in the National Institute of Allergy and Infectious Diseases trial, the primary endpoint was time to recuperation, which was indicated as recovered enough to leave hospital or a resumption of normal activity level. Currently, the ClinicalTrials.gov database registers 30 trials for remdesivir (6 trials are using the term "GS-5734'" instead of remdesivir) concerning COVID-19 [47]. However, more time is needed to evaluate all the data and information from these trials. Currently, three clinical trials are enrolling patients to study the therapeutic usefulness of remdesivir. One trial is organized in Hôpital Cochin in Paris, France, and two are in the USA. One clinical trial organized in Wuhan, China, was suspended [48].

## 8. Conclusions

In conclusion, it is accepted that the COVID-19 pandemic took the world and medical community by surprise. Currently, significant effort is directed towards the development of suitable therapeutic agents for this infection, be it a new vaccine or various therapeutics. Remdesivir as such represents one of the more promising alternatives for COVID-19 therapy, but the current understanding of this disease and of possible ways of dealing with it requires further investigation. In the future, studies will be necessary to look into the interactions of antiviral therapy (represented, i.e., by remdesivir) with other therapies that are being investigated for COVID-19. These are, for example, non-steroidal anti-inflammatory drugs (NSAIDs). So far, conclusive data for the use of NSAIDs in the treatment of COVID-19 patients are not available [49]. The investigation of the use of anti-inflammatory therapies includes various agents, such as glucocorticoids, IL-6 antagonists, JAK inhibitors, and others, as their

use may be (or maybe not) beneficial in COVID-19 patients with an impaired immune system [50]. Lately, the Janus kinase-signal transducer and activator of transcription (JAK-STAT) pathway is attracting the attention of clinicians as a potential target for successful immunomodulation [51]. However, also here the data obtained are conflicting and are not yet fully conclusive and need to be verified [52–54]. This is true for all COVID-19 therapies, as more time is needed for evaluating clinical experience and outcomes.

**Funding:** This research received no external funding.

**Acknowledgments:** The authors are grateful to Kuwait University for support.

**Conflicts of Interest:** The authors declare no conflicts of interest.

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
