# Peer review of "Remdesivir—Bringing Hope for COVID-19 Treatment"

_scipharm, doi:10.3390/scipharm88020029_

Round 1

Reviewer 1 Report

The manuscript “Remdesivir – bringing hope for COVID-19 treatment” concerns a very actually and urgent issue of the effective and safe COVID-19 therapy.

The structure of the manuscript is very good. However, there are a several points that need correction:

  1. I suggest that an adverse effects, which are listed in the Section 5 („Basic pharmacological data of remdesivir and its adverse effects in patients”) should be presented in a Table form.
  2. A short subsection within the Section 7 („Therapeutic uses of Remdesivir”) about the in vitro efficacy (on human cell types) and in vivo models (human and animal diseases) should be introduced.
  3. The paragraph about the possibility of use remdesivir in a combination with another agent (e.g. Janus kinase inhibitor) should be added.
  4. There is some typing errors, e.g.: page 6 line 235 lack of bracket.

After providing these remarks, this paper will be suitable for pubication in „Scientia Pharmaceutica”.

Author Response

  1. I suggest that an adverse effects, which are listed in the Section 5 („Basic pharmacological data of remdesivir and its adverse effects in patients”) should be presented in a Table form. We are not sure whether adding a table would help clarity. We did not do it but if the reviewer insists, we would oblige.
  2. A short subsection within the Section 7 („Therapeutic uses of Remdesivir”) about the in vitro efficacy (on human cell types) and in vivo models (human and animal diseases) should be introduced. The subsection was added as requested.
  3. The paragraph about the possibility of use remdesivir in a combination with another agent (e.g. Janus kinase inhibitor) should be added. Some therapeutic options, including Janus kinase inhibitors, are mentioned in the Conclusion.
  4. There are some typing errors, e.g.: page 6 line 235 lack of bracket. Corrected as requested by referee

The authors would like to thank the reviewer for his effort in evaluating our manuscript and for the recommendations.

Reviewer 2 Report

Manuscript titled “Remdesivir – bringing hope for COVID-19 treatment” describes remdesivir (GS-5734) as the promising therapeutic agent in COVID-19. The Authors presented mechanism of action for remdesivir, pharmacological data, adverse effects in patients, analytical determination of remdesivir and its metabolites. The most interesting and current for today is part concerning clinical trials evaluating remdesivir as a treatment for COVID-19 and therapeutic uses of remdesivir in other RNA infections, such as Ebola, Nipah virus infection, and Middle East Respiratory Syndrome (MERS). Manuscript is well organized and can be interesting for a wide range of researchers in the field of medicinal chemistry.

The main criticism concerns the following aspects:

In Figure 1 structure of adenosine 5’-phosphate should be placed in order to explain differences in structures between remdesivir and adenosine 5’-phosphate concerning adenine part. Three key structures: adenosine 5’-phosphate, remdesivir, and GS-441524 should be placed in Figure 1.

The term “1’-cyanoadenosine analog” is misleading, because this suggest that adenosine part is modified by 1’-cyano group. In fact adenine part is also changed. Remdesivir is adenosine triphosphate analog.

v.73, 106, 107, 164, and other – “…active metabolite 1’-cyanoadenosine triphosphate….” is incorrect, because adenosine (adenine) part is changed and name adenosine should not be used.

In chapter 3 the mechanism of activation of remdesivir as a prodrug into active metabolite triphosphate could be described more precisely (via the actions of esterases and a phosphoamidase and then nucleoside-phosphate kinases).

Additional comments:

Abstract, v.10-17 – too long sentence

Introduction, v.25-27 – two sentences are exact repetition from abstract

Chapter 3 – citation [4] is overused

Figure 1. - The structure of remdesivir in Figure 1 is too large.

Figure 2. – a triphosphate derivative is not a remdesivir metabolite No GS-441524; compound number GS-441524 is nucleoside analog according to [4]

v.119 – title should be better “6. Analytical determinations of remdesivir and its metabolite GS-441524”

v.130 - “Avataneo and her colleagues….” – citation [24] should be inserted in this place

v.135 – “The optimum separation was achieved by using a mobile phase was consisting of (H2O + formic acid 0.05%)…….” Should be: The optimum separation was achieved by using a mobile phase consisting of (H2O + formic acid 0.05%)……

v.211 – “Similarly to COVID-19 infection, this virus also causes respiratory and neurological disorders with 40 to 75% 211 mortality according to the World Health Organization [34]” – suggests that COVID-19 also causes mortality of 40 to 75%.

v.285 – “Currently, ClinicalTrials.gov database registers 24 clinical trials for remdesivir and 6 trials for GS-5734 concerning COVID-19 [44]”. The number of clinical trials should be summarized, because remdesivir and GS-5734 are synonyms.

Author Response

  • In Figure 1 structure of adenosine 5’-phosphate should be placed in order to explain differences in structures between remdesivir and adenosine 5’-phosphate concerning adenine part. Three key structures: adenosine 5’-phosphate, remdesivir, and GS-441524 should be placed in Figure 1.

Answer: Fig. 1 was modified according to the suggestion.

  • The term “1’-cyanoadenosine analog” is misleading, because this suggest that adenosine part is modified by 1’-cyano group. In fact adenine part is also changed. Remdesivir is adenosine triphosphate analog.

Answer: You are correct. We clarified this.

  • 73, 106, 107, 164, and other – “…active metabolite 1’-cyanoadenosine triphosphate….” is incorrect, because adenosine (adenine) part is changed and name adenosine should not be used.

Answer: You are correct. We clarified this.

  • In chapter 3 the mechanism of activation of remdesivir as a prodrug into active metabolite triphosphate could be described more precisely (via the actions of esterases and a phosphoamidase and then nucleoside-phosphate kinases).

Answer: The process of remdesivir activation was described more in detail.

Additional comments:

  • Abstract, v.10-17 – too long sentence

Answer: Corrected as suggested.

  • Introduction, v.25-27 – two sentences are exact repetition from abstract

Answer: The text was changed in the Introduction.

  • Chapter 3 – citation [4] is overused

Answer: The frequency of citing reference [4] was decreased.

  • Figure 1. - The structure of remdesivir in Figure 1 is too large.

Answer: The authors agree with the referee that the final appearance of Fig. 1 should be smaller in the final version. However, Fig. 1 was changed/modified according to the other recommendation.

  • Figure 2. – a triphosphate derivative is not a remdesivir metabolite No GS-441524; compound number GS-441524 is nucleoside analog according to [4]

Answer: The reviewer is correct. Consequently, we made the following change in Fig. 2: Remdesivir metabolite GS-441524 was changed to the following:

Remdesivir metabolite: triphosphate of nucleoside GS-441524

  • 119 – title should be better “6. Analytical determinations of remdesivir and its metabolite GS-441524”

Answer: Corrected.

  • 130 - “Avataneo and her colleagues….” – citation [24] should be inserted in this place

Answer: Citation 24 inserted.

  • 135 – “The optimum separation was achieved by using a mobile phase was consisting of (H2O + formic acid 0.05%)…….” Should be: The optimum separation was achieved by using a mobile phase consisting of (H2O + formic acid 0.05%)……

Answer: Corrected.

  • 211 – “Similarly to COVID-19 infection, this virus also causes respiratory and neurological disorders with 40 to 75% 211 mortality according to the World Health Organization [34]” – suggests that COVID-19 also causes mortality of 40 to 75%.

Answer: Clarified in the text that it is the Nipah virus causing the mortality.

  • 285 – "Currently, ClinicalTrials.gov database registers 24 clinical trials for remdesivir and 6 trials for GS-5734 concerning COVID-19 [44]". The number of clinical trials should be summarized because remdesivir and GS-5734 are synonyms.

Answer: Modified according to the suggestion.

The authors would like to thank the reviewer for his effort in evaluating our manuscript and for the excellent recommendations.